# Silver-Decorated Silicon Nanostructures: Fabrication and Characterization of Nanoscale Terraces as an Efficient SERS-Active Substrate

**DOI:** 10.3390/ijms24010106

**Published:** 2022-12-21

**Authors:** Mohammad Kamal Hossain

**Affiliations:** Interdisciplinary Research Center for Renewable Energy and Power Systems (IRC-REPS), Research Institute, King Fahd University of Petroleum & Minerals (KFUPM), Dhahran 31261, Saudi Arabia; kamalhossain@kfupm.edu.sa; Tel.: +966-13-860-1058; Fax: +966-13-860-7312

**Keywords:** silicon nanostructure, silver nanoparticle, surface-enhanced Raman scattering (SERS), SERS-active hotspots, finite difference time domain

## Abstract

Rich and highly dense surface-enhanced Raman (SERS) hotspots available in the SERS-active platform are highly anticipated in SERS measurements. In this work, conventional silicon wafer was treated to have wide exposure to terraces available within the silicon nanostructures (Si-NSs). High-resolution field emission scanning electron microscopic (FESEM) investigations confirmed that the terraces were several microns wide and spread over different steps. These terraces were further decorated with silver nanoparticles (Ag-NPs) of different shapes and sizes to achieve SERS-active hotspots. Based on more than 150 events, a histogram of the size distribution of Ag-NPs indicated a relatively narrow size distribution, 29.64 ± 4.66 nm. The coverage density was estimated to be ~4 × 10^10^ cm^−2^. The SERS-activity of Ag-NPs -decorated Si-NSs was found to be enhanced with reference to those obtained in pristine Si-NSs. Finite difference time domain models were developed to support experimental observations in view of electromagnetic (EM) near-field distributions. Three archetype models; (i) dimer of same constituent Ag-NPs, (ii) dimer of different constituent Ag-NPs, and (iii) linear trimer of different constituent Ag-NPs were developed. EM near-field distributions were extracted at different incident polarizations. Si-NSs are well-known to facilitate light confinement, and such confinement can be cascaded within different Ag-NPs-decorated terraces of Si-NSs.

## 1. Introduction

Surface-enhanced Raman scattering (SERS) is a non-invasive and non-destructive analytical detection technique that facilitates the identification of target analytes without exogenous labels [1]. The technique has evolved into a multidisciplinary research area encompassing diverse areas of interest, including fast and accurate detection of biomolecules, antibiotics, food additives, and single molecules [2,3,4,5,6,7,8,9,10]. Although the technique was found exciting initially, it was not useful enough in trace detection until the discovery of extraordinary enhancement in Raman cross-sections [11,12,13,14]. The cross-section of Raman scattering could be enhanced up to 10^14^ in the event of so-called “SERS-active hotspots (SHSs).” In the early 1980s, many pioneers, including R.P. Van Duyne, Jeanmaire, and M. Moskovits, suggested in their studies that the molecules at the interstitials of a tiny interparticle gap, ~1 nm, exhibit unusually large Raman-cross section through the excitation of localized surface plasmons resonances (LSPRs) [12,14]. Since then, extensive research on single-molecule SERS (SM-SERS) and nano-engineering of SERS-active substrates based on such SHSs concepts have been focused on experimentally and theoretically [7,8,10,15,16,17,18,19]. However, such SHSs are mostly unpredictable and uncontrolled in the SERS substrates because the field enhancement is very sensitive to the nanogeometry of SHSs and the relative position of the molecules in the SHSs [19,20]. SHSs engineering research occupies and resides in the core part of SERS research, shading all the focus on the design of a SERS-active platform of abundant and well-controlled SHSs straddling from zero (0D)-, one (1D)-, two (2D)- to three-dimensional (3D) spaces [18,19,20,21,22]. In 0D and 1D spaces, SHSs are mostly well recognized theoretically and experimentally due to limited interstitials and LSPRs, whereas in 2D space, the number of SHSs increases, and LSPRs excitations intermingle through hybridizations and coalescences [20,22,23,24,25]. Multi-dimensional plasmonic platforms have been reported using the 3D concept of SHSs arrangement wherein SHSs spread over x-, y- and z-planes and SHSs intensity boost substantially due to extensive plasmonic coupling within the laser excitation volumes [19,20,26,27,28,29,30]. To this extent, cascaded multiscale electromagnetic (EM) field enhancement due to the incorporation of terraces in elongated and plasmonic nanogaps increases the number of SHSs available on the entire substrate [27,31,32].

Qualitative SHSs, as well as a higher density of the same support, achieve reproducible and reliable SERS signals of a target analyte. Therefore, it is necessary to devise a plasmonic platform that will hold a multitude of SHSs spanning different planes. In order to achieve abundant and rich SHSs, the SERS fraternity has reported numerous studies, including tuneable plasmonic gaps, arrays of holes, tips, arbitrary spikes, and nanoparticles array [33,34,35,36,37,38]. It is well acknowledged that the strongest confinement can be achieved in a tiny gap between two plasmonic surfaces. However, it has been a great challenging task to control the growth and geometry of such plasmonic nanostructures with sufficient accuracy. In this context, silicon nanostructures (Si-NSs) were reported to be one of the promising nanostructures due to their high surface areas, high densities, high roughness, and high concentrations of the characteristic tips [39,40,41,42]. In addition, the Si-NSs are known to be 3D structures that exhibit high light scattering leading to an increased number of the sites of electric field confinements and thus enhancing the ensemble SERS signals [43,44]. With reference to 2D SERS substrates such as nanoholes array or nanoassemblies, the 3D plasmonic platform offers a higher surface-to-volume ratio leading to a larger specific surface area and, thus, higher density of SHSs achieved upon appropriate decoration. These structures were reported to have enhancement factors (EFs) higher than the 2D array due to the increased number of SHSs in a 3D structure. Fabrication of high-performance 3D SERS-active substrates has been reported by Song and his group. Vertically oriented Multistack nanogaps were prepared on individual nanopillars [45]. Bai et al. reported that SERS substrates consisting of a dense array of silicon nanowires (Si-NWs) decorated by silver nanoparticles (Ag-NPs) provided EFs of the order of 10^8^ [46]. Silicon nanorods decorated by gold nanoparticles have been reported by Lin and the group. Although the procedure to achieve such SERS-active substrate was complicated, the EF was achieved as high as 10^7^ [47]. Likewise, an ordered array of silicon nanorods decorated by Ag-NPs showed EFs as high as 10^6^ [48]. Recently, a SERS-active silver dendrite network was developed on silicon, and the sensitive detection of lysozyme with EF of 2.4 × 10^6^ was reported [49]. SERS-active silicon nanopillars decorated with silver were reported by Michael and his group. A self-leaning mechanism was demonstrated to achieve a large number of SHSs, and thus higher and uniform EF was reported [50].

Here in this work, Si-NSs were fabricated, followed by further treatment to have a wide exposure of terraces in different steps. Henceforth, terraces provided higher surface area as well as enough space to possess highly populated SHSs. The terraces were several microns wide and extended over several stages, according to high-resolution field emission scanning electron microscopic (FESEM) investigation. These terraces were decorated with Ag-NPs of different shapes and sizes. Based on more than 150 events, a histogram of the size distribution of Ag-NPs, as well as coverage density, was estimated. SERS-activity of the Si-NSs and Ag-NPs-decorated Si-NSs was carried out using Rhodamine 6G (R6G, C_28_H_31_N_2_O_3_Cl) as standard Raman active dye. When compared to pristine Si-NSs, the SERS activity of Ag-NPs-decorated Si-NSs was observed to be increased. EM near-field distributions derived from finite difference time domain (FDTD) validate the observations. EM near-field distributions at different incidence polarizations were extracted using FDTD analysis using three archetypal models: a dimer of the same constituent Ag-NPs, a dimer of different constituent Ag-NPs, and a linear trimer of different constituent Ag-NPs. The parameters of the FDTD model geometries were set so that the model accurately reflects the experimental observation as validated by FESEM. Light confinement is well-known in Si-NSs, and this confinement may be cascaded across various Ag-NPs-decorated Si-NS terraces. Such an approach to designing rich and abundant SHSs opens up new avenues to achieve efficient nanoplasmonics-based electronic and optoelectronic devices.

## 2. Results and Discussion

### 2.1. Topographic Confirmation

Understanding the nanoscale geometry of a plasmonic SERS-active platform with utmost accuracy is indispensable for SERS study. It has been reported previously that a small change in the local geometry of SHSs leads to a huge degree of variation in inducing EM near-field distributions. In this context, the as-fabricated Si-NSs and Ag-NPs-decorated Si-NSs were thoroughly investigated by high-resolution FESEM. Figure 1a shows a FESEM micrograph of the Si-NSs. A zoom-in view of the same image has been further investigated, as shown in Figure 1b. Figure 1b represents a magnified view (20 µm × 20 µm) as marked by the white dotted square in Figure 1a. The as-fabricated Si-NSs were found to be full of microscale valleys of different sizes and openings, as shown by a 3D hawk-eye in Figure 1c. A line profile along the white dotted line, as shown in Figure 1b, is shown in Figure 1d. The dips and hills and the distances between such dips and hills clarified further that the microscale valleys were wide and open. Within such valleys, a multitude of surface areas were speculated to be available in a small area that can hold abundant SHSs upon decoration. Indeed, terraces of wide and open exposure within the Si-NSs were observed, as shown in Figure 1e. Figure 1e represents a high-resolution FESEM micrograph of the same NSs consisting of terraces at different levels in the nanoscale. Further magnified view of the same Si-NSs revealed terraces of micrometer length. Figure 1f shows a zoom-in view of the area (5 µm × 5 µm) as marked by the white dashed square in Figure 1e. Terraces of different areas were observed, as shown in Figure 1f. Approximate lengths of typical terraces were estimated following a line profile, as shown in Figure 1g. Figure 1g displays a line profile along the white dotted line as indicated in Figure 1f. Two typical terraces of ~1.68 µm and ~2.24 µm were marked therein by double-sided arrows in Figure 1f. As stated earlier, such terraces can be decorated with appropriate metal nanoparticles, and strong enhancement in the SERS signal can be achieved.

Figure 2a shows a FESEM micrograph of Ag-NPs-decorated terraces available in Si-NSs. A zoom-in view, as shown in Figure 2b, represents a magnified view (20 µm × 20 µm), as marked by the white dashed square in Figure 2a. Similar to that observed in pristine Si-NSs, Ag-NPs-decorated Si-NSs were found to be full of microscale valleys of different sizes and openings, as confirmed by a 3D hawk-eye view, as shown in Figure 2c. A line profile along the white dotted line, as shown in Figure 2b, is displayed in Figure 2d. The dips and hills and the distances between such dips and hills clarified further that the microscale valleys were wide and open. Valleys of different sizes and shapes were observed, as shown in Figure 2e. Figure 2e represents a high-resolution FESEM micrograph of the same Ag-NPs-decorated Si-NSs consisting of terraces in different steps in microscale valleys. A further magnified view of the same Ag-NPs-decorated Si-NSs revealed terraces of micrometer length. Figure 2f shows a zoom-in view of the area (5 µm × 5 µm) as marked by the white dashed square in Figure 2e. Terraces of different areas were observed, as shown in Figure 2f. Approximate lengths of typical terraces were estimated following a line profile, as shown in Figure 2g. Figure 2g displays a line profile along the white dotted line as indicated in Figure 2f. Two typical terraces of ~1.47 µm and ~0.82 µm were marked therein by double-sided arrows in Figure 2f. Such terraces of micrometer lengths can have millions of NPs at the top, although the FESEM micrographs were not able to clearly depict such a scenario in this regard. In this context, high-resolution FESEM investigations with sufficient high magnification indicated the existence of Ag-NPs available in Ag-NPs-decorated terraces of Si-NSs.

Figure 3a shows a high-resolution FESEM micrograph of Ag-NPs-decorated terraces of Si-NSs. It is noteworthy that the surface of the Si-NSs was decorated with Ag-NPs of different sizes and shapes. Figure 3b represents a magnified view (500 nm × 500 nm) as marked by the white dotted square in Figure 3a. A line profile along the white dotted line in Figure 3b is shown in Figure 3d. Four typical Ag-NPs, as marked by “1”, “2”, “3,” and “4” in Figure 3b, were estimated to be ~27.8, ~47.2, ~22.2, and ~50.1 nm, respectively, as shown in Figure 3d. High-resolution FESEM micrograph also supported that the Ag-NPs used to decorate Si-NSs were indeed of different sizes and shapes. Figure 3c shows a zoom-in view (500 nm × 500 nm) of the white dashed square, as marked in Figure 3a. Three typical Ag-NPs of ~16, ~38, and ~75 nm in diameter were shown by a white dotted circle in Figure 3c. Based on more than 150 events as observed in a high-resolution FESEM micrograph, a size distribution was estimated. Figure 3e shows a histogram of the size distribution of Ag-NPs along with the Gaussian fit in red. As shown in Figure 3e, a relatively narrow size distribution, 29.64 ± 4.66 nm using the Gaussian fit (in red) was estimated. The narrow full width at half maximum (FWHM) of 4.66 nm inferred that as-fabricated Ag-NPs were mostly uniform in size. As stated earlier, Ag-NPs of different sizes and shapes facilitate higher EM near-field distribution, and thus enhanced Raman signal is expected. At the same time, the population of the higher EM near-field distribution enriches the quality of the ensemble SERS enhancement.

### 2.2. SERS-Activity

In SERS activity, the target analyte needs to be as close as possible to SHSs [51,52,53]. SHSs facilitate EM near-field confinements by two to five orders. Interestingly, this enhanced EM near-field supported both excitation and emitted radiation to be energized, leading to SERS intensity as per the fourth power of field enhancement. The detailed mechanism, known as two-fold enhancement, has been explained in the later part of the text. The “lightning rod” effect makes 1D-nanostructures more advantageous in localizing EM near-field distributions at two distinct places, whereas isolated spherical plasmonic nanoparticles have smaller EM near-field distributions. However, as compared to isolated nanoparticles or nanorods, 2D assemblies have been shown to have more EM near-field distribution locations. On the contrary, 3D nanostructures such as Si-NSs provide even higher specific areas to possess SHSs in different planes. Therefore, it is important to have a higher population of SERS-active interstitials (synonymously known as SHSs), as was demonstrated in Ag-NPs-decorated terraces of Si-NSs in Figure 3 and Figure 4.

Figure 4a shows the SERS spectrum of R6G adsorbed on Ag-NPs-decorated Si-NSs excited at 633 nm. Ten SERS bands of R6G were noted, as shown in the inset of Figure 4a. The inset of Figure 4a represents a selected area of the spectrum as marked by the black dashed rectangle. The SERS bands of R6G, as observed in this case, were marked by the red vertical dashed lines therein. SERS spectrum of R6G molecules acquired from Ag-NPs-decorated Si-NSs exhibited strong intensities at 610, 769, 1120, 1182, 1310, 1361, 1508, 1571, 1595, and 1648 cm^−1^ wavenumbers as marked by red vertical dashed lines in Figure 4a. As stated in Table 1, 610, 769, 1120, 1182, and 1310 cm^−1^ wavenumbers represent C−C ring bending mode (in-plane) in phenyl rings, C−H bending mode (out-of-plane), C-H bending mode (in-plane) xanthene/phenyl rings, C−H bending mode (in-plane) in xanthene ring, C-C Hybrid stretching mode in xanthene/phenyl rings and NHC_2_H_5_ groups, respectively. Bands at 1361, 1508, 1571, 1595, and 1648 cm^−1^ wavenumbers correspond to the aromatic stretching vibration modes.

Figure 4b displays a CCD image of the same specimen with a white “x” indicating the focusing point. Due to the modest intensity and exposure of laser excitation, no damage or dissociation of R6G dyes was observed. The laser was switched off immediately after the measurement, as previously stated, without moving the focusing spot on the same object. The true color CCD image revealed the possible sites of plasmon-active Ag aggregates, as shown in Figure 4d. Figure 4d shows zoom-in views of a small area marked by the white dashed square in Figure 4b along with four typical sites “1”, “2”, “3,” and “4”. The color variation of the CCD image indicated that Ag aggregates were different in nature and thus speculated to induce EM near-field differently. Figure 4c displays the Raman spectrum of R6G of 0.2 M adsorbed on Si-NSs only. Table 1 lists the SERS bands observed during this investigation, as well as the band assignments that corresponded to reported SERS peaks of R6G [51,52,53]. As previously stated, the EM enhancement mechanism is the most important aspect of SERS enhancement; hence it was hypothesized that EM near-field dispersion in Ag-NPs-decorated Si-NSs aided increased SERS enhancement. As noted earlier in the experimental section, extensive simulations were carried out to extract EM near-field distributions for three specific Ag-NPs models.

SERS EF of R6G adsorbed on Ag-NPs-decorated Si-NSs has been estimated as shown below. It is to be noted that ten SERS bands of R6G were observed, therefore EF was estimated and compared using the following formula,
(1)$$EF=\left(\frac{I_{SERS}}{I_{bulk}}\right)\times \left(\frac{C_{bulk}}{C_{SERS}}\right)$$
where $I_{SERS}$, $I_{bulk}$, $C_{SERS}$, and $C_{bulk}$ represent the intensity of the SERS peak, the intensity of the Raman peak, the molarity used in SERS, and the molarity used in Raman measurements at a specific vibrational mode, respectively.

As shown in Table 1, the SERS enhancements for R6G at the band of 610, 769, 1120, 1182, 1310, 1361, 1508, 1571, 1595, and 1648 cm^−1^ wavenumbers were estimated to be ~1.71 × 10^6^, ~1.17 × 10^6^, ~3.70 × 10^6^, ~7.20 × 10^5^, ~3.93 × 10^6^, ~8.08 × 10^5^, ~4.28 × 10^6^, ~7.55 × 10^5^, ~4.36 × 10^5^, and ~3.81 × 10^5^, respectively. It was noteworthy that Si double tone mode was observed at ~933–996 cm^−1^.

### 2.3. EM Near-Field Distributions

In SERS enhancement, there is a contribution from the charge transfer (CT) mechanism in most of the scenarios. However, the EM mechanism is known to be dominant and several orders higher, particularly for metal-based plasmonic SERS-active substrates. Pittinger et al. proposed the EM mechanism evidenced by twofold EM enhancement in 1986, and it has since become the basic mechanism of EM enhancement in the SERS process. In brief, the two-fold EM enhancement happens in two subsequent processes, as shown in equation 2. In the first process, EM near-field enhances Raman scattering of the analyte, whereas the scattered Raman signal gets enhanced further in the second [53,54,55,56,57]. The EM-based EF is denoted by
(2)$$EF={\left|\frac{E_{L}\left(\lambda_{I}\right)}{E_{I}\left(\lambda_{I}\right)}\right|}^{2}\times {\left|\frac{E_{L}\left(\lambda_{I}\pm \lambda_{R}\right)}{E_{I}\left(\lambda_{I}\pm \lambda_{R}\right)}\right|}^{2}=EF_{1}\left(\lambda_{I}\right)\times EF_{2}\left(\lambda_{I}\pm \lambda_{R}\right)$$
where $E_{I}$, $E_{L}$, $\lambda_{I},$ $+\lambda_{R}\;$, $-\lambda_{R}\;$, *EF*_1_ and *EF*_2_ are the incident electric field, local electric field, the excitation wavelength, anti-Stokes wavelengths and Stokes wavelength, first EF, and second EF, respectively.

Now we are in the position to elucidate EM near-field distributions extracted from FDTD analysis for archetype models, as shown in Figure 5. Three typical models; (i) a dimer of Ag-NPs of 30 nm diameter each, (ii) a dimer of Ag-NPs of 40 and 20 nm diameter and (iii) a trimer of Ag-NPs of 40, 20, and 30 nm diameter, were designed, and simulated for s-, p- and 45° of incident polarizations. Excitation of 633 nm that was normal to the geometries was used as per the experimental conditions. It is noteworthy that for a specific interstitial, EM near-field distribution follows the incident polarization direction. Although horizontal, vertical, and 45° (oblique) of incident polarizations have been demonstrated in this study, incident polarization angle of 10° interval can be found elsewhere [22,23,51].

According to high-resolution FESEM observations, as mentioned earlier in Figure 2 and Figure 3, the average diameter of Ag-NPs was estimated to be 29.64 ± 4.66 nm. Therefore, geometries of trimer and dimer models were chosen in three combinations: (1) a dimer of Ag-NPs of 30 nm diameter each, (2) a dimer of Ag-NPs of 40 and 20 nm diameters and (3) a trimer of Ag-NPs of 40, 20, and 30 nm diameters. Figure 5a represents EM near-field distributions of the trimer at XY (Z = 0) plane for the model geometry, as shown in inset (i) of Figure 5a, and excited with incident excitation of s-polarization. EM near-field distributions with a maximum intensity of 12.42 V/m were found to confine at the interstitials of the trimer. It was noteworthy that the adjacent NPs in model geometry were 1 nm apart, and therefore, EM near-field was localized at the junction at the excitation of s-polarization. In the case of a dimer of Ag-NPs of 30 and 20 nm diameters, as shown in the inset (ii) of Figure 5a, the EM near-field was found to confine at the interstitial with low intensity, E_max_ = 9.206 V/m compared to that observed in the trimer. On the other hand, for the dimer of Ag-NPs of 30 nm diameter each, as shown in inset (iii), the EM near-field confined at the interstitial was found to confine at the interstitial with nearly similar intensity, E_max_ = 9.206 V/m to that observed in the case of the dimer of Ag-NPs of 30 and 20 nm diameters. Insets (ii) and (iii) of Figure 5a represent the EM near-field distributions at s-polarization for the model dimers, as shown in inset (i) of Figure 5b and Figure 5c, respectively.

In the case of p-polarization, all three models showed no confinement at the interstitials since the interparticle axes were out of plane, as shown in Figure 5b. Figure 5b represents EM near-field distributions of the trimer at XY (Z = 0) plane for the model geometry, as shown in inset (i) of Figure 5a, and excited with incident excitation of p-polarization. EM near-field distributions with a maximum intensity of 3.042 V/m were found to confine at the surface of individual NPs. Insets (ii) and (iii) of Figure 5b represent the EM near-field distributions at p-polarization for the model dimers as shown in inset (i) of Figure 5b and Figure 5c, respectively. No confinement was observed in both scenarios, and the maximum electrical fields, E_max_ = 3.064 V/m and E_max_ = 2.756 V/m were found to confine at the surface of the dimers as shown in insets (ii) and (iii), respectively. However, in the case of incident polarization of 45°, all the models showed reasonably high EM near-field distribution, as shown in Figure 5c. Figure 5c represents EM near-field distributions of the trimer at XY (Z = 0) plane for the model geometry as shown in inset (i) of Figure 5a and excited with incident excitation of oblique (45°)-polarization. EM near-field distributions with a maximum intensity of 8.788 V/m were found to confine at the interstitials of the trimer. Although the interparticle axis was not in-plane to the incident polarization, EM near-field was localized at the junction at the oblique (34°)-polarization. In the case of dimer of Ag-NPs of 30 and 20 nm diameters as shown in the inset (ii) of Figure 5c, the EM near-field was found to confine at the interstitial with low intensity, E_max_ = 6.513 V/m compared to that observed in the trimer. On the other hand, for the dimer of Ag-NPs of 30 nm diameter each, as shown in inset (iii), the EM near-field confined at the interstitial was found to confine at the interstitial with nearly similar intensity, E_max_ = 6.816 V/m to that observed in the case of a dimer of Ag-NPs of 30 and 20 nm diameters. Insets (ii) and (iii) of Figure 5c represent the EM near-field distributions at the oblique (34°)-polarization for the model dimers as shown in inset (i) of Figure 5b and Figure 5c, respectively. FDTD simulations revealed that the trimer had stronger EM near-field distribution compared to those extracted for dimers of different geometries, as shown in Figure 5a–c. Figure 5d displays a bar graph of the maximum EM near-fields obtained for three models at s-, p-, and 45° of incident polarizations. As stated earlier, the trimer exhibits higher EM near-field distributions at s- and oblique (45°) of incident polarization compared to that of dimers. On the other hand, at p- polarization, all three models showed uniform EM near-field distributions. The inset of Figure 5d shows scattering cross-sections obtained at s (horizontal)-, p (vertical)- and 45° (oblique) of incident polarizations using all the three models used in FDTD simulation. In the case of the trimer, the scattering cross-sections at s-, p-, and 45° of incident polarizations were found to be higher compared to those in the case of dimers.

## 3. Materials and Methods

### 3.1. Si-NSs and Ag-NPs-Decorated Terraces in Si-NSs

Metal-catalyzed electroless etching is one of the popular and well-reported techniques to etch Si and achieve Si-NSs. As stated above, there are extensive studies in this regard, and almost all of the reports tried to achieve vertically aligned Si-NWs. However, as a SERS-active substrate, the underlying platform requires a large number of SHSs, and in such a scenario, vertically aligned Si-NWs were not able to support the needful. The inherent features of a metal catalyst, particularly shapes and sizes, play crucial roles in defining the nanostructures during the etching of silicon [39]. Therefore, metal catalysts of arbitrary shapes and low gap distribution were achieved to fabricate Si-NSs that consisted of wide and open valleys. A detailed topographic investigation was carried out and explained in the later part of the text.

A fabrication schematic is shown in Figure 6. Silver nitrate (AgNO_3_ 99.8% from HACH, Colorado, United States), hydrogen peroxide (H_2_O_2_ 30% in H_2_O from Scharlau, Barcelona, Spain), sulfuric acid (H_2_SO_4_ 98% from Applichem Panreac, Barcelona, Spain), and hydrofluoric acid (HF 48% from Applichem Panreac, Barcelona, Spain) were used as received. The silicon wafer was treated and used as base material to achieve the Si-NSs [58]. The wafer was catalyzed in an aqueous solution of HF/AgNO_3_ (5M/0.02N) for 1 min, as depicted in the catalyzing step in Figure 6. As shown in the etching step in Figure 6, the wafer was immersed in an etching solution of HF/H_2_O_2_ (5M/30%) for 15 min. An aqua-regia acid solution in water, H_2_O:HCl:HNO_3,_ was used to remove the residual Ag from the etched Si-NSs. Finally, the wafer was transferred to a vacuum chamber for thermal evaporation (model # Quorum K975X, East Sussex, UK) of Ag on the Si-NSs, as shown in the thermal evaporation step in Figure 6. The wafer was kept 10 cm away from the source, and a tungsten basket was used to evaporate Ag wire of 10 mg at a current of 10 A. Topographic confirmation, and further detailed investigations of Si-NSs and Ag-NPs-decorated Si-NSs were carried out using high-resolution FESEM (model # Tescan LYRA3, Kohoutovice, Czech Republic). Recently we have reported decoration of 2D nanoassemblies of gold nanoparticles with silver mists [18]. It was noted that mist-like silver clusters of different sizes and shapes could be developed through the thermal evaporation technique. Therefore, a similar technique was adopted to deposit Ag-NPs on the terraces or valleys of Si-NSs. In this investigation, no attempt was made to control the sizes of Ag-NPs. It is well understood that by increasing the amount of evaporating materials, the deposition eventually would turn into a continuous film.

### 3.2. SERS Set-Up

SERS measurements were carried out using LabRAM HR Evolution Raman microspectrometer (Horiba, Vénissieux, France) in the range of 30–4000 cm^−1^. Detailed SERS setup can be found elsewhere [59]. In brief, HeNe laser kits of 633 nm (17 mW) were used as an excitation source. The intensity of the laser was adjusted to 50% to avoid sample damage, and the same power was maintained for all the measurements. A long working distance lens (50×) was used to focus the laser and collect the SERS signal in a backscattering configuration. The exposure time and accumulation time for all samples were kept at 10 sec. and 2, respectively, at a grating of 600 gr/mm. A standard Raman-active dye R6G (Chroma GesellschaftSchmid GMBH & Co., Stuttgart, Germany) was used as received. As-fabricated Si-NSs and Ag-NPs-decorated Si-NSs were incubated by R6G dye of 0.2 M and 1 × 10^−6^ M, respectively, for ca. 10 min. The sample was washed with deionized water copiously before the SERS measurements. Raman and SERS measurements were repeated 3 to 4 times at each specific site of interest.

### 3.3. FDTD Simulation

As stated earlier, EM near-field distribution is acknowledged to be the main ingredient for giant SERS enhancement. Therefore, it is imperative to explore EM near-field distributions for specific SERS-active platforms. In this context, three specific models were developed for FDTD analysis, and EM near-field distributions were analyzed using a Maxwell equations solver, PLANC-FDTD (Information and Mathematical Science Laboratory Inc., Ver. 6.2, Tokyo, Japan). In the first model, the dimer of Ag-NPs of 30 nm diameter each was used and EM near-field distributions were analyzed for s-, p- and 45° of incident polarizations. In the second model, the dimer of Ag-NPs of 40 and 20 nm diameter each was developed, and EM near-field distributions were extracted for s-, p-, and 45° of incident polarizations. In the third model, a linear trimer of Ag-NPs of 40, 20, and 30 nm diameter each was designed, and EM near-field distributions were demonstrated for s-, p-, and 45° of incident polarizations. Although the constituent Ag-NPs were distinct from each other, notably in size and form, as revealed in FESEM examinations, nanoobjects were deemed smooth and ordered in a periodic pattern for the purpose of simplicity.

## 4. Conclusions

Si-NSs were fabricated using a metal-catalyzed electroless etching process and then treated to increase the number of terraces exposed inside the Si-NSs as-fabricated. Terraces provided more surface area as well as sufficient room for densely crowded SHSs. The terraces were several microns broad and extended over several stages, according to high-resolution FESEM investigations. Ag-NPs of various shapes and sizes were used to embellish these terraces. A histogram of the size distribution of Ag-NPs was calculated to be ~29.64 ± 4.66 nm. The coverage density was estimated to be ~4 × 10^10^ cm^−^^2^. SERS-activity of the pristine Si-NSs and Ag-NPs-decorated Si-NSs was carried out. In the presence of Ag-NPs-decorated Si-NSs, ten prominent SERS bands of R6G have been observed. The SERS-activity of Ag-NPs-decorated Si-NSs was found to be enhanced with reference to those obtained in pristine Si-NSs. The scenario was supported by EM near-field distributions extracted from FDTD analysis. EM near-field distributions at different incidence polarizations were extracted using FDTD analysis for three archetypal models: (i) a dimer of the same constituent Ag-NPs, (ii) a dimer of different constituent Ag-NPs, and (iii) a linear trimer of different constituent Ag-NPs. The maximum EM intensity was found in a trimer of Ag-NPs of various shapes, with E_max_ = 12.42 V/m at s-polarization and both interstitials as SHSs. In comparison to p-polarization, oblique polarization of input excitation resulted in a relatively high EM intensity, E_max_ = 8.788 V/m. Light confinement is well-acknowledged for Si-NSs, and this confinement may be cascaded over various Ag-NPs-decorated Si-NS terraces.

## Figures and Tables

**Figure 1 ijms-24-00106-f001:**
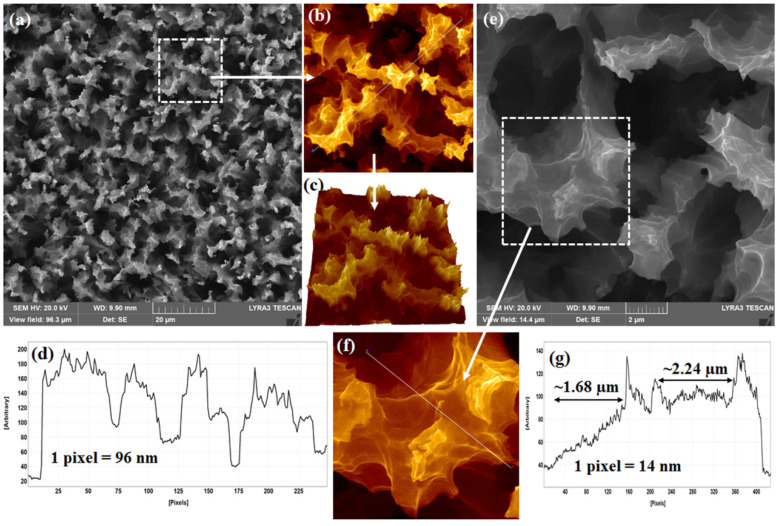
(**a**) FESEM micrograph of as-fabricated Si-NSs, (**b**) Zoom-in view of a small area (20 µm × 20 µm) as marked by the white dotted square in Figure 1a, (**c**) Hawk-eye view (3D) of the same confirming terraces of various sizes available in Si-NSs, (**d**) Line profile along the white dotted line crossing as-fabricated Si-NSs as shown in Figure 1b confirming the hills and valleys that contain different dimensions of terraces, (**e**) High-resolution FESEM micrograph of as-fabricated Si-NSs, (**f**) Zoom-in view of a small area (5 µm × 5 µm) as marked by the white dotted square in (**e**), and (**g**) Line profile along the white dotted line crossing as-fabricated Si-NSs as shown in (**f**) confirming the variation in sizes and shapes of terraces.

**Figure 2 ijms-24-00106-f002:**
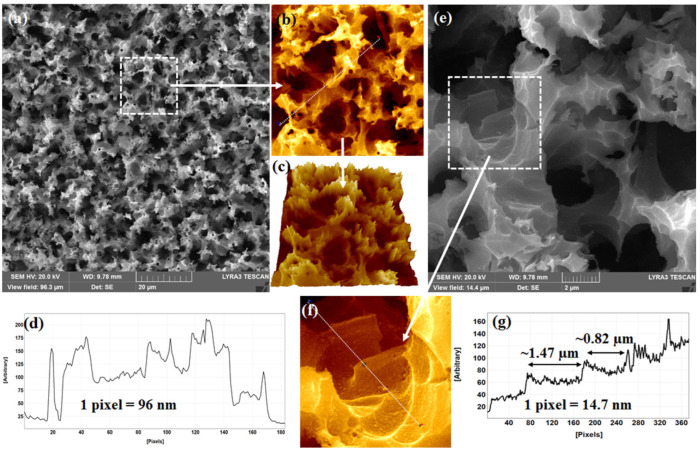
(**a**) FESEM micrograph of Ag-NPs-decorated terraces in Si-NSs, (**b**) Zoom-in view of a small area (20 µm × 20 µm) as marked by the white dotted square in (**a**), (**c**) Hawk-eye view (3D) of the same confirming terraces of various sizes available in Ag-NPs-decorated terraces in Si-NSs, (**d**) Line profile along the white dotted line crossing Ag-NPs-decorated terraces in Si-NSs as shown in (**b**) confirming the hills and valleys that contain different dimensions of terraces, (**e**) High-resolution FESEM micrograph of Ag-NPs-decorated terraces in Si-NSs, (**f**) Zoom-in view of a small area (5 µm × 5 µm) as marked by the white dotted square in (**e**), and (**g**) Line profile along the white dotted line crossing Ag-NPs-decorated terraces in Si-NSs as shown in (**f**) confirming the variation in sizes and shapes of terraces.

**Figure 3 ijms-24-00106-f003:**
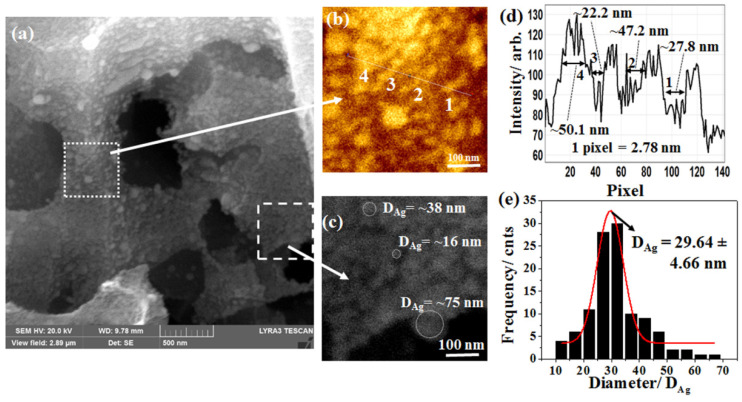
(**a**) High-resolution FESEM micrograph of Ag-decorated terraces in Si-NSs, (**b**) Zoom-in view of a small area (500 nm × 500 nm) as marked by the white dotted square in (**a**), (**c**) Zoom-in view of another small area (500 nm × 500 nm) as marked by the white dotted square in Figure 3a confirming different sizes and shapes of Ag-NPs, (**d**) Line profile along the white dotted line crossing four typical Ag-NPs marked as 1, 2, 3 and 4 in (**b**), and (**e**) size distribution (black) of Ag-NPs along with Gaussian fit (red).

**Figure 4 ijms-24-00106-f004:**
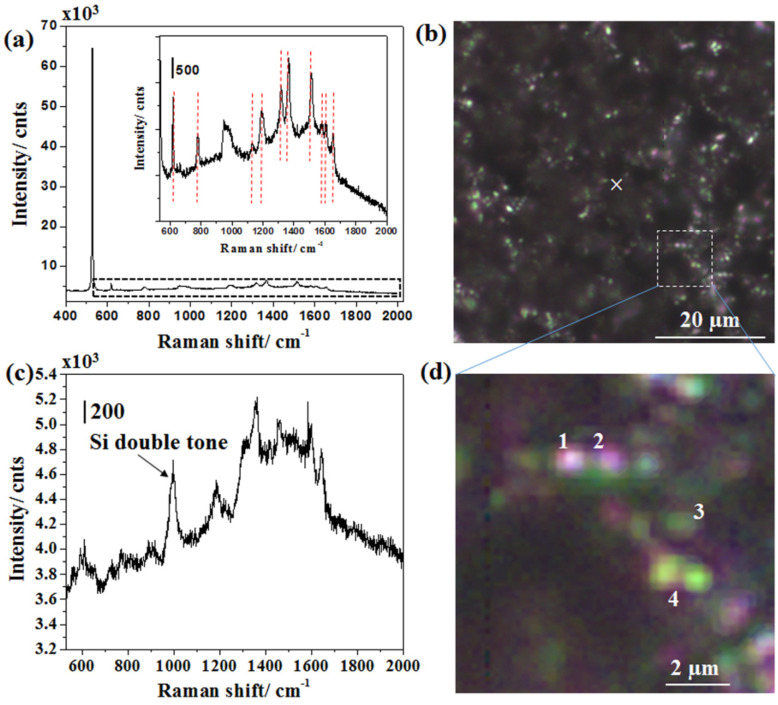
(**a**) SERS spectrum of R6G adsorbed on Ag-NPs-decorated Si-NSs; inset: Magnified view of the spectrum marked by a black dashed rectangle therein. Vertical dotted red lines in the inset represent SERS bands of R6G. (**b**) Bright-field microscopic image of the same indicating the spot of interest in acquiring SERS spectrum marked as the white “x” therein, (**c**) Raman spectrum of R6G adsorbed on as-fabricated Si-NSs and (**d**) Zoom-in view of a small area of interest as marked by a black dashed square in (**c**). Four typical sites of diffraction limited Ag-NPs decoration are marked by 1, 2, 3 and 4 in (**d**).

**Figure 5 ijms-24-00106-f005:**
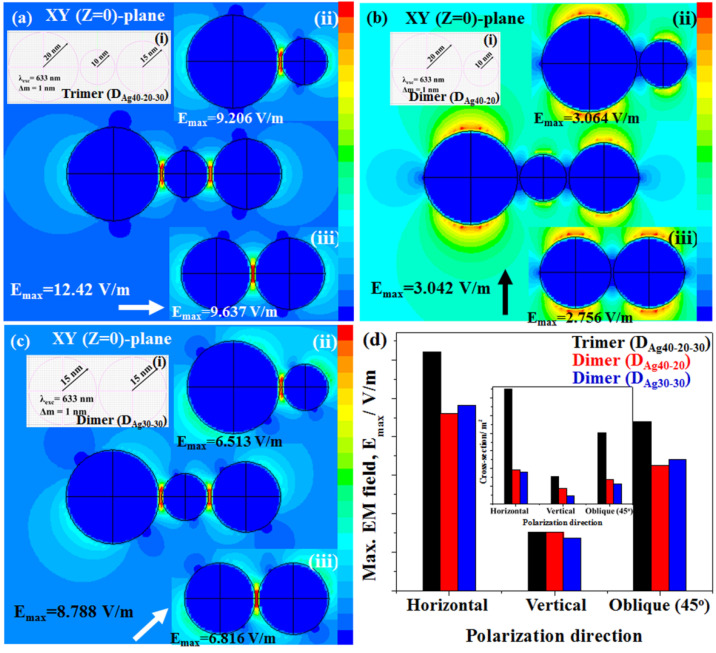
(**a**–**c**) EM near-field distribution of a typical Ag-NPs-based trimer model geometry excited at s-, p- and 45° of incident polarizations, respectively; inset (i): Corresponding model geometry along with simulation parameters and inset (ii) and (iii): EM near-field distribution of the dimer of same NPs size and that of different NPs size, respectively. Corresponding arrows and color bars represent the polarization directions and respective intensities observed under the simulations and (**d**) Maximum EM near-field of trimer and dimers simulated at s-, p- and 45° of incident polarizations; inset: Bar graph of forward cross-sections as obtained in FDTD simulations of trimer and dimers at s-, p- and 45° of incident polarizations.

**Figure 6 ijms-24-00106-f006:**
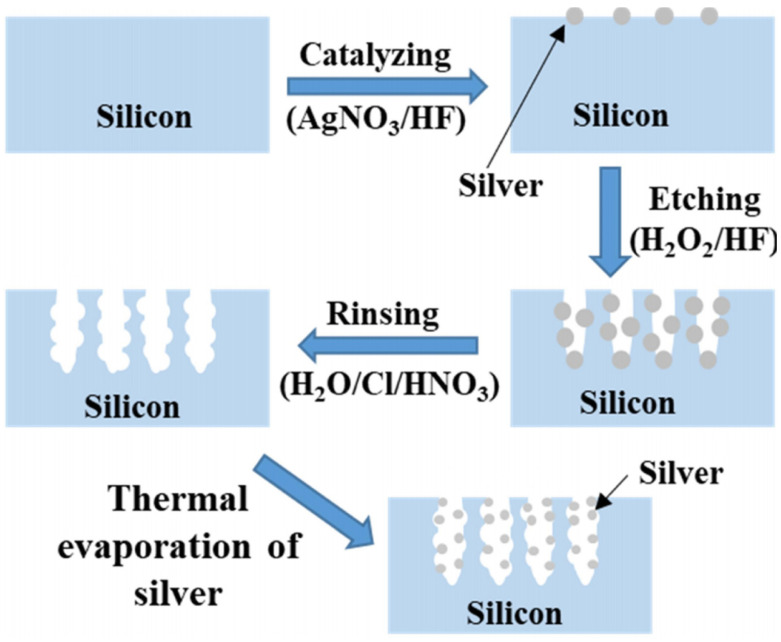
Free-hand schematic depicting the involved steps (i.e., catalyzing, etching, rinsing, and evaporation) in achieving Ag-decorated terraces in Si-NSs from a clean Si wafer.

**Table 1 ijms-24-00106-t001:** SERS bands of R6G, corresponding band assignments, and estimated EF for individual bands in the presence of Ag-NPs-decorated Si-NSs.

SERS Band of R6G (cm^−1^)	Band Assignments	EF
610	C−C ring bending (in-plane) in phenyl rings	~1.71 × 10^6^
769	C−H bending (out-of-plane)	~1.17 × 10^6^
1120	C-H bending (in plane) xanthene/phenyl rings	~3.70 × 10^6^
1182	C−H bending (in-plane) in xanthene ring	~7.20 × 10^5^
1310	C-C Hybrid stretching in xanthene/phenyl rings and NHC_2_H_5_ group	~3.93 × 10^6^
1361	C−C stretching in xanthene ring	~8.08 × 10^5^
1508	C−C stretching in xanthene ring	~4.28 × 10^6^
1571	C-C stretching in phenyl ring	~7.55 × 10^5^
1595	C−C stretching in phenyl ring	~4.36 × 10^5^
1648	C−C stretching in xanthene ring	~3.81 × 10^5^

## Data Availability

Not applicable.

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
