# Peer review of "Silver-Decorated Silicon Nanostructures: Fabrication and Characterization of Nanoscale Terraces as an Efficient SERS-Active Substrate"

_ijms, 2022, doi:10.3390/ijms24010106_

Round 1

Reviewer 1 Report

Abstract:

You should put the large name of EM before writting the acronyms. The same in the keywords with FDTD. It is written the large name of all, but not in the case of FDTD.

Introduction:

include the next references with the references 2-10, after "food additives and single-molecules: 

"Nanostructured hybrid surface enhancement Raman scattering substrate for the rapid determination of sulfapyridine in milk samples." 10.1016/j.talanta.2018.10.047

"Analytical nanometrological approach for screening and confirmation of titanium dioxide nano/micro-particles in sugary samples based on Raman spectroscopy-capillary electrophoresis." 10.1016/j.aca.2018.10.067

"Analytical control of Rhodamine B by SERS using reduced graphene decorated with copper selenide." 10.1016/j.saa.2019.117302

Materiales and methods:

Pag. 5 line 130: change the comma to "and"

How many times have the experiments been done? What is the reproducibility of the method? The authors say that they obtain nanoparticles of a different size and shape. How is this controlled? Don't you get different sizes and shapes? If it hasn't been done, it should be done. If it has been done, it should be indicated in the text.

SERS-set-up:

Pag. 6 line 153:  what does it mean ca. 10 mins?

What happens if the concentration of Rhodamine 6G is increased or decreased? Why was this concentration chosen?

Topographic confirmation:

Fig 2e and Fig 3e do not have the same work distance.

I can´t see in Fig. 4b the numbers 1,2,3 and 4.

In the work, the authors worked with horizontal, vertical and 45 (oblique). What will happen if you work with another angle?

Finally, what is the next step with this paper? Do you think about an application? What do you think you can do with your material? It will be good if you write what do you want to do with them.

Reviewer 2 Report

The author describes the fabrication of SERS active substrates using a straigh-forward approach for black-si and PVD. Nevertheless, some question may arise while reading this paper and should be answered:

1. The fabrication of the Si-NWs uses an electroless approach as described in fig 1. Why does the author etch the silver nanoparticles and then add silver nanoparticles. What is the SERS performance using the already deposited Ag-NP instead of the subsquent fabricated Ag-NP?

2. The Ag-NP histogram analysis is not sufficient. The Ag-NP deviation might not be defined by its real-size. The size deviation might by influenced by the limited resolution of the measurement setup. Can the author comment on that please?

3. What is the limit of detection (LOD) of R6G for the Si-NWs and the Ag-NP decorated Si-NWs shown in this study?

4. Did the author measure the life-time of the Ag-NP decorated Si-NWs as silver might degrade?

5. The structures, investigated in this paper, reveal a large surface. Could hydrophobicity be a factor for a lower detectable EF?

Reviewer 3 Report

This manuscript is reporting on the fabrication of nanostructured silicon decorated with Ag-NPs for SERS application. Although, this manuscript could be of interest to the readers of this journal. There are a few issues that need to be addressed before suggesting for publication.

-          The introduction needs to be strengthened by providing more references as to why nanostructured silicon is an ideal substrate for SERS and biomedical application. Here are a few examples doi.org/10.1021/acsami.8b10590 and doi.org/10.1021/acsami.6b14836

-          Each panel in the figures needs to be mentioned/explained in the corresponding caption and the text.

-          The fonts in the figures are not legible.

-          Could the author include the EF equation used to calculate the EF.

-          The limit of detection should be measured and reported

-          Could the author compare the SERS activity of nanostructured Silicon before and after decorated with Ag-NPs?

-          It is recommend to use more than only one dye to measure the EF

Round 2

Reviewer 1 Report

Thank you for revising and including the comments I sent. I think the paper is ready to bu published.